# Gender-Based Dating Violence and Social Media among Spanish Young People: A Qualitative Study

**DOI:** 10.3390/bs14070575

**Published:** 2024-07-07

**Authors:** Marta García-Ruiz, María Dolores Ruiz-Fernández, María del Mar Jiménez-Lasserrotte, Isabel María Fernández-Medina, María Isabel Ventura-Miranda

**Affiliations:** 1Virgen de las Nieves University Hospital, 18014 Granada, Spain; marta.garcia.ruiz@hotmail.com; 2Department of Nursing Science, Physiotherapy and Medicine, University of Almeria, 04120 Almeria, Spain; mjl095@ual.es (M.d.M.J.-L.); ifm731@ual.es (I.M.F.-M.); mvm737@ual.es (M.I.V.-M.); 3Facultad de Ciencias de la Salud, Universidad Autónoma de Chile, Temuco 4780000, Chile

**Keywords:** dating violence, young people, social media, university students, perceptions

## Abstract

(1) Gender-based dating violence is common among adolescents. This violence has global repercussions and can have immediate and delayed consequences on health. Also, cases of dating violence and sexual abuse using technology are increasing. The aim of this research is to describe and understand the perceptions and experiences of Spanish university students aged 18 to 22, about gender-based dating violence and its perpetuation through social media. (2) A qualitative descriptive study was used, following the five consolidated criteria for reporting and publishing COREQ qualitative research. (3) The inductive analysis of the data obtained in the focus group session and the individual interviews of the twelve participants was organised into three major themes: the concept of gender violence that Spanish youth have, the education they have received on gender-based violence and whether they consider that social media are a way to exercise this type of violence. (4) Spanish youth have a broad vision of the attitudes and behaviours that make up gender-based dating violence in an affective relationship. The education received at home is of vital importance for young people, but not all receive it. Social media are frequent tools through which many young people perpetuate controlling partner violence and normalise aspects and situations of gender violence, making it necessary to stress them in prevention programs.

## 1. Introduction

According to the World Health Organization [1], youth is defined as people between the ages of 10 and 24, with adolescence being defined as the period between the ages of 10 and 19. During this period, romantic relationships can be stressful and overwhelming due to the restructuring of interpersonal roles, issues around identity formation and sexuality and possible rejection and break-ups that challenge conflict resolution and coping skills [2]. 

The denomination of these romantic relationships, or courtship, varies according to different authors. On the one hand, some of them consider courtship as a formal relationship between two people that will end in marriage, while others consider it as a relationship between a couple that is known by family and friends but does not have a marital purpose [3]. Also, courtship could be defined as a period of acquaintance between both partners who are attracted both physically and mentally, perform social activities together and have a marital or non-marital purpose [4]. However, courtship has evolved over time, along with society. Until the beginning of the 20th century, the union between two people was far from the romantic character it has today in the West, still perpetuated in some cultures, consisting of an arranged union based on economic, religious, class or political interests for families [5]. Accordingly, interactions between the two partners have changed over the years. Until the 19th century, romantic dates took place in the intimacy of marriage and family, extending to other public spaces at the end of this century, distancing them from family life [6]. Currently, young couples live their romantic relationships through mobile phones and social media, which are the ways they use to communicate [7]. Both mobile devices and social media influence the behaviour and attitudes of young people, and therefore their courtships, becoming a reflection of today’s society and generating both positive and negative changes in relationships [8].

Within courtship, we find dating violence (DV) which refers to any emotional, physical violence, sexual abuse, stalking or cyber abuse [9] that comes from a couple in which at least one of the partners is young [10]. This type of violence is common among adolescents, with the highest prevalence rate found in young people aged 18–22 [10]. In the case of Spain, up to 46.6% of women aged 16–24 years had experienced some type of violence by their partner [11]. Due to a higher incidence in women [12], the fact that this type of violence tends to have worse consequences for the female gender [13] and that research on gender-based violence has shown that these attitudes begin in courtship [14], it was considered important to analyse this type of violence with a gender-based approach (gender-based dating violence), emphasising that gender-based violence refers to “any act of violence that results in threatened or actual physical, sexual or psychological harm, including threats, coercion or arbitrary deprivation of liberty, whether occurring in public or in private life” ([15], p.2). A study conducted in Andalusian universities found that the most common behaviours within GBDV were, firstly, cyberbullying (68.22%), secondly, psycho-emotional violence (49.71%), followed by control and surveillance attitudes (44.63%), sexual abuse (16.68%) and, finally, physical violence (5.60%) [16]. Another research study conducted in Canada stated that one in three young people were victims of any physical, psychological and/or cyberbullying in the last year [17]. Similarly, a study conducted in Brazil stated that 31.9% of adolescent participants had experienced physical violence, 36.4% sexual violence and 81.8% psychological violence [18]. This violence has global repercussions [19] and can have immediate and delayed health consequences [13]. These consequences include mental health problems such as depression, post-traumatic stress disorder and/or anxiety [20,21] or physical health problems such as unplanned pregnancies or STDs [22]. In addition, academic problems, suicidal ideation and even suicide attempts may occur [23].

Over time, GBDV has been analysed in a similar way to adult violence, summarised as sexual, physical, psychological and bullying violence, with a gap forming in the study of other types of GBDV such as abuses via social media [24], such as Facebook (Meta Platforms Inc., Menlo Park, CA, USA), Instagram (Meta Platforms Inc., Menlo Park, CA, USA) and X (previously Twitter) (X Corp., San Francisco, CA, USA), which have enormous social influence, constituting the main means of entertainment and news dissemination [25]. Although there is a fair amount of research on in-person (face-to-face) forms of GBDV, how it can occur via technology is barely being assessed [26].

Cases of dating violence and sexual abuse through the use of technology are on the rise [10]. This type of violence is associated with poorer mental health, a higher risk of substance use, a higher risk of other forms of non-digital violence [27] and a higher number of suicide cases [28]. 

On the other hand, knowing the perceptions of this type of violence is of vital importance in order to prevent it, since these beliefs intervene in the interpretation of violent behaviour. However, no attention has been paid to these perceptions [22].

Therefore, the aim of this study was to describe and understand the perceptions and experiences of young Spaniards aged 18–22 on gender-based dating violence and its perpetuation through social media.

## 2. Materials and Methods

A descriptive qualitative study was carried out as it provides a description of subjective ideas within a sample of people on a given topic [29]. Furthermore, this research modality was chosen as it uses systematic criteria that allow for the organisation of the study’s phenomena to be established, providing systematic information that is comparable with other studies [30]. The consolidated criteria for reporting and publishing qualitative research COREQ [31] were followed.

Participants were young people aged between 18 and 22 with Spanish nationality. They were recruited through purposive sampling via the social media Instagram and Twitter during the month of January 2023, following the following inclusion criteria: being between 18 and 22 years old (both inclusive), residing in Spain and being a person who cooperates. Finally, 12 young people participated whose characteristics and demographic data are reflected in Table 1. As can be seen, all the participants had a university education and some of them combined their studies with temporary jobs. Most of the young people were female and their sexual orientation was predominantly heterosexual. Furthermore, only one young person in the sample was transgender (a person who is born with a biological sex with which they do not identify, perceiving themselves as a different sex, without considering a physical change necessary) [32]. On the other hand, most of the sample was in a courtship, whether formal or informal. Finally, for a better understanding of the table, we described some terms such as cis, a Latin word meaning “on the same side of”. Therefore, cisgender people were those who identify with the same gender they were assigned at birth due to their biological sex [33]. On the other hand, the term “closed couple” refers to a romantic relationship between two people who have mutually agreed not to engage in sexual or affective romance with other people. Finally, the expression “roll” refers to an affective and/or sexual relationship between two young people that they do not consider as serious or formal, but as temporary and of little relevance.

Data collection was carried out through one focus group session and twelve individual interviews via video calls using the Meet platform, which lasted approximately 60 min and between 20 and 45 min, respectively. A script of questions (Appendix A) was used to answer the study questions. 

The focus group session was conducted in February 2023 by the author of this paper, with no one else present, who acted as the moderator. The individual interviews were conducted in February and March 2023. Their content was recorded with a recording device for later transcription. Data collection ended when no new information was provided when analysing the data, as data saturation had been reached. 

Prior to conducting the focus group session and interviews, socio-demographic data were collected, the protocol was explained (Table 2), data confidentiality was guaranteed and signed informed consent was obtained.

The analysis of the data obtained was carried out using ATLAS ti.23. The focus group session and the individual interviews were transcribed, and notes were taken on the non-verbal language of the participants. 

After transcription, the data were analysed according to the steps described by Braun and Clarke [34]: (1) familiarisation with the data: after transcription of the data, the researcher read the transcripts in order to reach a general understanding of what was described by the participants; (2) initial code generation: representative quotes were chosen and assigned interesting feature codes using the “open coding” and “in vivo coding” function in ATLAS ti.23; (3) initial theme generation from data coding: initial themes were generated by grouping codes with similar meanings that were linked through a central idea; (4) theme development and review: a consensus on themes and sub-themes was reached after independently verifying that all generated themes were consistent with the codes and quotes. A table was created with examples of the coding strategy using significant quotes, sub-themes and topics (Table 3); (5) refining, defining and naming issues: the report was refined by selecting the essential parts; and (6) writing the report: during writing, the most illustrative quotes were selected, and summaries of eloquent examples were elaborated on, and the analysis was linked back to the research question and the literature.

The report was carried out in accordance with the ethical principles of the Declaration of Helsinki. A certificate was obtained from the ethics committee of the University of Almeria, registration number: EFM 231/23. 

Participants were informed of the aim of the study, the voluntary nature of participation and the possibility of withdrawing from the study at any time. Permission to record the focus group session and the interviews was also requested and informed consent was obtained from all participants prior to the start of data collection. The anonymity of the data was preserved by replacing the names of the participants with a code included in Table 1 consisting of the letter P and a consecutive number (P1, P2…).

The duty of confidentiality and anonymity was ensured in accordance with Organic Law 3/2018, of 5 December, on the Protection of Personal Data and Guarantee of Digital Rights. Digital rights were guaranteed in compliance with the general regulation of Regulation 2016/679 of the European Parliament and of the Council of 27 April 2016.

To assess the scientific rigour of the study, Guba and Lincoln’s [35] quality criteria were adopted: credibility, as the data collection process was reproduced in detail and the analysis process was reviewed by independent researchers; transferability, as there was a thorough description of the participants, their context and the sampling method; dependability, as the interpretations were reviewed by an external researcher who was not involved in the data collection; and confirmability, as the transcripts were returned to the participants to verify their accuracy.

## 3. Results

Twelve young Spanish nationals aged 19–22 participated in this study. After the inductive analysis of the data obtained from the perceptions and experiences of the participants, supported by the focus group session and the interviews of the participants, the results were organised around three main themes: the extent of gender-based dating violence according to young people, the education received about gender-based dating violence and social media and the ease of engagement in gender-based dating violence.

On an interpretative/pragmatic level of analysis, a conceptual map linking the different categories to each other was developed (Figure 1). 

In this figure, we can see that the central axis of the study, gender-based dating violence, is influenced by different aspects. On the one hand, it can manifest itself as physical, sexual or psychological violence, the latter being perpetuated more frequently through insults, control and influence over the victim by the abuser, emotional blackmail and invalidation and invisibility as a person, as well as attempts to create guilt in the victim. 

On the other hand, it can be observed that aspects such as insecurity, self-esteem/self-love and jealousy, as well as the education that young people have received in the different areas of their lives (high school, home, university), influence this type of violence.

Likewise, social media are a means through which to perpetuate gender-based dating violence through activities such as sexting, but they can also be a good way to spread education against this type of violence and therefore have both positive and negative connotations for the young participants. 

Finally, the participants have pointed out throughout the different aspects that encompass gender-based dating violence the existing differences between genders.

### 3.1. The Extent of Gender-Based Dating Violence According to Young People

The participants of our study considered that gender-based dating violence (GBDV) could be exercised through different types of abuse such as physical, psychological and sexual violence. They emphasise that physical violence such as assault or pushing and shoving is the most serious. Manipulative attitudes or psychological violence were pointed out as the beginning of an abusive relationship and the most present in most toxic relationships. Sexual violence was the least mentioned in our research, being included in physical abuse.

Some of the participants were of the opinion that this violence is influenced by the gender of the abuser and the victim and is more often perpetrated by men against women. On the other hand, some of the interviewees did not discriminate between genders, speaking of violence perpetrated by one person against another. 

*“I think that gender violence occurs at the moment when the partner interferes with the rights and freedoms of his partner, both physically and psychologically”*. (P4)

Seven of the twelve participants indicated that they had experienced some form of gender-based dating violence from their partners in their current or previous relationships. Some of them used the term “toxic attitudes”. Raising the tone of speech, controlling attitudes and insults were present in their relationships. These behaviours made the victim feel insecure and guilty. As a result, they reported trying to change their actions to avoid arguments and confrontations. 

*“The other day I took a photo of myself and he says ‘fuck, how sexy for the boys to see’. It makes me realise that it looks like I’m putting it up for the boys to see. And yes, there have also been more extreme things like a push, basically”*. (P9)

*“Yes, I was with an ex-partner who was very aggressive or instead of talking things through, he immediately resorted to shouting, fighting… physically he never did anything to me”*. (P8)

Several of the participants stated that they found a link between acts of gender-based dating violence in partners and feelings such as insecurity and jealousy in the abuser, as well as low self-esteem and self-respect in the victim. They considered that some of the controlling behaviours stemmed from the abuser’s insecurities towards herself, feeling jealous of the victim’s friends or other boys with whom she might associate. They also point out that a person with high self-esteem and positive self-worth is less likely to be locked into an abusive relationship and more likely to be able to leave it.

*“In the end it’s up to her to gain a bit more self-esteem and self-respect because I think that in the end many problems stem from that. So, I think it would be very important to go to a psychologist for her to see what she is worth and from there to gain confidence and manage to leave that relationship”*. (P9)

*“My previous partner became very jealous, very distrustful, and in the end, this creates an insecurity in you too about anything you are going to do, which could make him feel bad and lead to a fight”*. (P3)

#### The Scourge of Psychological Violence in Courtship

Psychological violence was the aspect most present and repeated by the participants. They pointed out that this type of abuse is often subtle and goes unnoticed, being less evident than physical or sexual aggression. Different concepts such as coercion, control, invalidating, belittling or blaming were mentioned by almost all the interviewees in this research. 

Participants identified that abusers limited their partners’ freedom and self-expression by controlling their friendships, the clothes they wore, where they went and other decisions they made. They also pointed out that abusers’ aggressive behaviour and the use of high tones and insults were intended to devalue their victims and make them feel inferior.

*“Some kind of conflict would happen and it was always my fault or I would manage to turn things around so that I would feel guilty”*. (P8)

*“It’s that I think that many times society tends to see violence as very extreme cases, but we don’t realise that other comments like the one from the group of friends, that I don’t like them…. and something like that, so simple, that it seems that he says it for other kinds of things, because in the end it is still considered violence for me”*. (P9)

### 3.2. The Education Received on Gender-Based Dating Violence 

The education that the participants had received throughout their lives about gender-based dating violence took place in different settings. As a result, three sub-themes emerged from this main theme: education received at high school, university and education by parents at home.

#### 3.2.1. The Lack of Sex Education Received at School

Most of the participants reported having received some kind of talk during their school years. These took place in the first years of Compulsory Secondary Education (ESO). However, they were less frequent in the last years of the Baccalaureate, being displaced in importance by the entrance exam, even though this is a stage in which many young people begin their first stable relationships. 

The lack of usefulness of these talks was emphasised by many participants. They claimed that they did not focus on gender-based dating violence, how to identify it and how to leave it. They dealt with topics such as contraception and drug or alcohol addictions. They also pointed out that they were given little importance by the teachers themselves, being something additional to the homework and the syllabus seen in class.

*“Yes, I think I remember that two girls came to the school a long time ago to talk to us about gender-based violence and so on, but I think they came more about drugs, alcohol, but about gender-based violence, sex education and all that, I don’t remember very much”*. (P8)

*“The talk at school didn’t help me at all. But also because of the way it was carried out and because it was not given importance”*. (P10)

#### 3.2.2. Differences in Sexuality Education at the University Level

At the university level, differences were found to exist between students in the health care field and students in other fields. 

Participants who study or have studied nursing, medicine or physiotherapy reported having received extensive GBDV education throughout their studies. Throughout their time at university, they had either core or elective subjects in which GBDV was emphasised. They also emphasised the importance of identifying potential victims and helping those who came to health centres or emergency rooms seeking assistance.

*“I have given a lot about gender violence, because even the teachers themselves have told us that it is a problem that is present in the issue of being a health worker, above all”*. (P10)


*“Yes, I am now in a subject that is gender and health, which is an optional subject. We talk a lot about sexuality, the definition of gender and sex and I think we are also going to deal with issues of gender violence. And it’s the one where we are going to talk about it the most”.*
(P8)

However, the participants who had studied other subjects such as teaching or history pointed out that the education they had received about GBDV was superficial. Furthermore, they emphasise that what they have seen has depended on the teachers who voluntarily added it to their subject. 

*“In the faculty it is true that we had a subject called Sociology, but it was not focused on gender or anything, but the teacher I had focused on that…. But it was because she wanted to”*. (P4)

#### 3.2.3. Sex Education in Each Household

The education received at home by parents was very different for each of the participants in this study. On the one hand, six of the twelve participants indicated that GBDV is not a topic that is talked about at home, or that it is treated as a taboo subject, thinking that it is something alien to their children.

*“Now maybe a bit more. But not before, because these situations were more… they were more normal and less talked about and the woman was blamed more and ‘she must have done this for something’”*. (P11)

On the other hand, half of the sample of participants did receive education about GBDV from their parents. Even so, they emphasise that they have not received a specific talk as such, but rather general comments on interpersonal relationships and individual values. 

*“And then in my house, they also talked a lot about “being very careful”, that is, being very careful that nobody disrespects you. To have your rules very clear and the moment someone crosses that barrier, such as, for example, disrespect, to say ‘this is it’. And don’t forgive them”*. (P6)

All the participants agree on the importance of education at home and not only in the school and/or university environment. They point out that this is an issue that is sometimes made invisible, especially in terms of psychological abuse, and which should be addressed, regardless of the age or education of each person. They point out that parents and society often take young people’s knowledge for granted and do not work on it. 

*“I think they should teach us a bit more, not only at school, but also to our own parents, because it is a taboo subject: “this will never happen to my daughter…” you never know”*. (P3)

### 3.3. Social Media: Ease of Engagement in Gender-Based Dating Violence

When inquiring about social media and their influence on the romantic relationships of 18–22-year-olds with Spanish nationality, participants pointed out both positive and negative aspects.

The negative aspect of social media in interpersonal relationships was more relevant and important for all participants in this research. They emphasised the control that the abuser can easily exert over the victim in terms of her friendships, the people she interacts with, what she does and where she is. They were also pointed out as a source for generating jealousy and insecurities in the abuser and a way to foster more dependent and toxic relationships. Participants also highlighted the idealisation of relationships that exists in the media, which generates frustration in couples.

*“As for the abusive partner, negative. Because there are more possibilities of certain things happening that I don’t like and, well, in relationships like that, they try to control the victim”*. (P10)

*“Uploading things to media, photos, many people talk to you… and that also generates more jealousy”*. (P9)

In terms of positive aspects, most participants agreed that social media make it easier and more convenient to stay in touch with your partner, especially in long-distance relationships. In addition, they point out that they can be a way to show the love and affection you feel for your partner, by showing off or uploading photos together. 

They also point out that if you are single, you can meet more people through social media. Likewise, in the case of the victim, they can make contact with other people who have gone through a similar situation and find support.


*“The fact that we can use the mobile phone to communicate with our partner, imagine in a long-distance relationship or, even if it is not long-distance, to be able to communicate…. I think it’s obviously very positive, it can strengthen the relationship”.*
(P9)

For almost all participants, social media would be a good tool for disseminating education on gender-based dating violence to both young and old, as they are part of our daily lives. They point out that it is a medium in which content can go viral and reach a large number of people, especially if it is shared by so-called influencers. They also point out that social media can be used as a way of entertaining and bringing talks given in schools closer to the younger generations.

*“Yes, because nowadays it is very present in young people and what they consume most are social media. So, if we start to educate in a good way through social media, I think it can have a positive influence”*. (P6)

Participants emphasised the differences between genders when it comes to gender-based dating violence through social media. They pointed out that the female gender tends to use this medium to reproach their partner’s attitudes, while the male gender exercises controlling behaviours through them, monitoring followers, likes and comments on their partner’s accounts.

*“I think it is independent of gender, both are the same, but the intention with which it is done is different. The female gender does it as if to investigate or even to recriminate, but the male gender does it more to prohibit and say you have to change this or that”*. (P10)

#### Sexting: A Dangerous Practice

Regarding the practice of “sexting” or sending erotic content/messages through social media, participants stressed the danger of losing control over the content sent, especially in the case of photos or videos. Therefore, they also pointed out the importance of educating young people about this. They also stressed that the consequences of disseminating this type of content on social media do not affect women and men in the same way. They considered that the male gender tends to request erotic material from the female gender, the latter being more susceptible to discrimination and humiliation in the case of the dissemination of the content. 

On the other hand, they stressed the importance of carrying out this practice only when there is sufficient trust in the couple and in the safest possible way, avoiding, for example, that screenshots can be taken. In these cases, they believe it is a beneficial and positive behaviour for the couple to encourage sexual interest and desire.

*“I see it as a good resource as long as it is with a person you trust, that you have established limits or that it doesn’t go beyond that. If you send photos and so on, then be very careful, try to send it in the most secure way or encrypt it so that there are no problems later on”*. (P3)

## 4. Discussion

The main objective of this study was to describe and understand the perceptions and experiences of young Spaniards aged 18–22 about gender-based dating violence and its perpetuation through social media. It has been observed that participants consider gender-based dating violence as a broad spectrum of abusive attitudes, including physical, sexual or psychological violence. In terms of the education they have received on gender-based dating violence throughout their lives, discordance was found among the participants. Some of them have received extensive education both at university and from their parents, while others have had a more ambiguous and scarce education. Finally, all participants were aware of the duality of social media, which can be used in a negative way to exert control in the relationship or in a positive way as a means of communication for the couple. 

Regarding the concept of gender-based dating violence, in the case of the research conducted by Marcos and Isidro [36], participants pointed to concepts such as physical and/or sexual force, coercion and emotional abuse among others. Furthermore, in a study conducted in the United States [37], young people interviewed pointed to the fact that society often thinks of GBDV as physical abuse. However, behaviours such as forcing your partner to do something they do not want to do, yelling at them and emotional and verbal abuse are also GBDV attitudes within dating. All of these terms coincide with the perceptions of the participants in our study. 

Likewise, in our research, young people identified jealousy as an important factor involved in partner GBDV attitudes. This perception coincides with the results of other studies in which participants stated that jealousy is one of the main causes of GBDV in couples [38]. Similarly, they identified jealousy as the main source of arguments and conflicts within adolescent couples [39]. On the other hand, students who participated in research conducted in Chile [40] affirmed that these GBDV behaviours can be perpetrated by both men and women in different settings such as school or work. These results coincide with the beliefs of the majority of our participants, who also pointed out these aspects throughout the focus group session and the interviews.

Furthermore, another aspect mentioned throughout the research is the belief held by some participants that the victim bears some responsibility for her partner’s violence against her, attributing this to low self-esteem. These misconceptions may contribute to the perpetuation of gender-based violence in intimate partner relationships, so it will be crucial to address them in the educational environment of young people.

With regard to the education received on GBDV in the university setting, our participants from the health field reported having received extensive training on gender-based dating violence throughout their degree, while students from other fields reported not having specific subjects and agendas on the subject. In contrast to our research, the results obtained in the study conducted by Diéguez, Martínez-Silva, Medrano and Rodríguez-Calvo [41] indicated that participants with health care degrees had not taken subjects on GBDV, unlike law and criminology students.

On the other hand, the education that young people receive about GBDV from their parents is of vital importance, since depending on the model of education used by parents, behavioural styles will be transmitted, which adolescents will reproduce in their personal relationships [42]. Likewise, young people who have a good family relationship, in which parents are present, are less likely to tolerate or perpetuate some type of intimate partner violence [13,43]. More frequent conversations between parents and children about the influence of, for example, social media on their well-being are important [10]. As observed in a study conducted in Seville, there is a greater likelihood of intimate partner violence in those young people who were brought up in a harsh and authoritarian way [42] compared to adolescents whose parents are more lenient [44].

Finally, regarding social media and the partner, in the previous study by Marcos and Isidro [33], the participants considered the control that could be exercised by WhatsApp and other social media to be GBDV, coinciding with the thinking of the participants in our research. Furthermore, on this violence of control exercised by young people through social media, different studies carried out in the Basque Country [45] and in Murcia [46] observed that 42% and 35.68% of their sample, respectively, had suffered this type of abuse by their partners. This result agrees with our participants, who pointed out that the main negative aspect of social media is the control that the abuser can exert over the victim, which is common nowadays. 

For all these reasons, preventive educational programmes to avoid or reduce the occurrence of gender-based dating violence are of vital importance for young people. However, some programmes implemented in the Spanish educational system have not been effective due to various reasons: they are based on different theories, they use instruments to measure violence without scientific rigour, the duration of the programme (number of sessions) or the lack of long-term evaluation to check the objectives achieved [47]. Fortunately, there are programmes that have proven their effectiveness, which are useful in combating this type of violence among young people. At the international level, the “Safe Dates” programme [48] analysed its impact four years later, without observing violent behaviour in the participants. In Spain, Hernando [49] developed a programme based on the analysis of films and documents that achieved changes in the attitudes and knowledge of young people aged 16–18. On the other hand, the “PREVIO” programme was able to benefit mainly young people who claimed to have perpetrated physical and verbal violence against their partners. Likewise, the role of parents and society in general is crucial to combat gender-based dating violence, encouraging young people to acquire values such as responsibility, personal freedom, solidarity, tolerance, equality, justice and respect [13].

As a limitation of the study, all participants were of Spanish nationality and had a university education. Also, most of the sample was female. Therefore, the participants’ responses may be influenced by the coping and educational patterns found in Western societies, by their level of education and by their gender. As future lines of research, similar studies could be conducted in other cultures and societies with a more equal sample in terms of gender and level of education, in order to compare the results obtained. 

## 5. Conclusions

Young Spaniards have a broad view of the attitudes and behaviours that make up gender-based dating violence. Although the level of knowledge learned about this type of violence at school and university has been diverse, in general, it is a topic that is dealt with from high school to the end of higher education. Education received at home is of vital importance for young people, but not all of them receive it. Social media are frequent tools through which many young people perpetuate controlling violence towards their partners. Despite being aware of them, most of them continue to suffer or carry it out.

Therefore, it highlights the importance of education not only in the educational sphere but also in the personal sphere for young people, so that they can recognise a situation of gender-based dating violence in time and, furthermore, not continue to engage in toxic behaviour and attitudes in their emotional relationships. Likewise, social media normalise aspects and situations of gender-based dating violence, and it is necessary to focus on them in prevention programmes.

## Figures and Tables

**Figure 1 behavsci-14-00575-f001:**
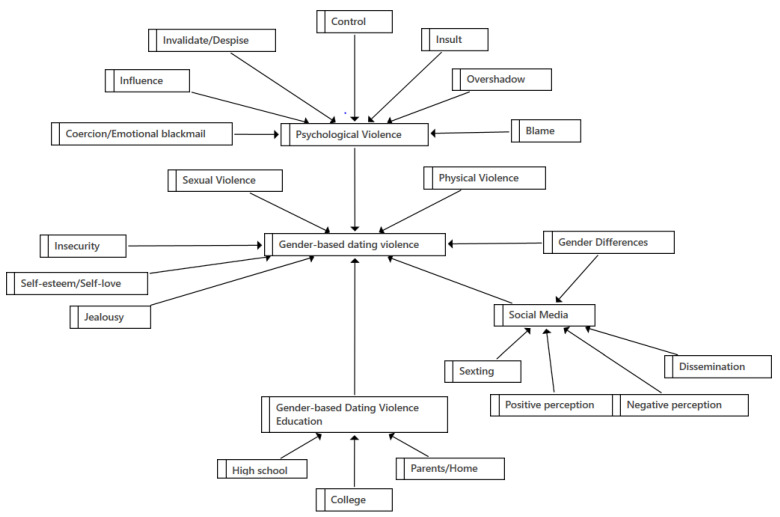
Conceptual map.

**Table 1 behavsci-14-00575-t001:** Participant demographics.

	Age	Gender	Sexual Orientation	Sentimental Situation	Years in Relation	Studies/Work
P1	22	Woman cis	Heterosexual	Closed, distant pair	2.5 years	Medicine and waitressing
P2	21	Men cis	Heterosexual	Single	-	Medicine and chess professor
P3	22	Woman cis	Bisexual	Roll	10 months	Physiotherapist and Master’s student
P4	22	Woman cis	Heterosexual	Roll	<1 months	Teacher and private tutor
P5	22	Woman cis	Heterosexual	Single	-	Medicine
P6	21	Woman cis	Homosexual	Single	-	Nursing
P7	22	Woman cis	Heterosexual	Closed couple	7 years	History
P8	19	Woman cis	Heterosexual	Closed couple	1 year	Nursing
P9	22	Woman cis	Heterosexual	Roll	4.5 years	Nursing
P10	21	Trans man	Bisexual	Closed couple	1.5 years	Nursing
P11	22	Woman cis	Heterosexual	Single	-	Social educator
P12	22	Woman cis	Heterosexual	Closed couple	3 years	Nursing

**Table 2 behavsci-14-00575-t002:** Focus group session and interview protocol.

**Presentation**	A reminder of why the study is being carried out, the purpose of recording the interviews and the focus group, the voluntariness and freedom of the participant at all times, etc.
**Sample questions on the concept of dating violence**	What do you understand to be gender-based dating violence behaviours? What education on gender-based dating violence have you received throughout your life?
**Examples of questions on social networking**	How do you consider that social media and new technologies influence a relationship? What do you think about certain behaviours such as checking your partner’s mobile phone, checking “likes” or monitoring your partner’s followers on their social media?
**Examples of questions on personal experiences of dating violence**	What are your experiences of gender-based dating violence either in your current relationship or in a previous relationship? If you think someone you know is being a victim of GBDV, how would you help them?
**Final question**	Do you want to comment on any other issue, or ask me any questions…?

**Table 3 behavsci-14-00575-t003:** Examples of themes, subthemes and units of meaning.

Themes	Subthemes	Units of meaning
The Extent of Gender-Based Dating Violence According to Young People	The Scourge of Psychological Violence in Courtship	Invisibilization, invalidate, belittle, insult, control, influence, coerce, emotional blackmail, blaming
The Education Received on Gender-Based Dating Violence	The Lack of Sex Education Received at School	Purpose, topic’ talk, importance, interest’ degree, duration, school stage
Social Media: Ease of Engagement in Gender-Based Dating Violence	Sexting: A dangerous Practice	Danger, confidence, care, turmoil, lack of education, couple’ benefits, gender’ difference

## Data Availability

The dataset is available upon request from the authors.

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
