# Peer review of "Gender-Based Dating Violence and Social Media among Spanish Young People: A Qualitative Study"

_behavsci, 2024, doi:10.3390/bs14070575_

Round 1

Reviewer 1 Report

Comments and Suggestions for Authors

This study is on an important topic.  There are word choice issues and inconsistencies throughout that need to be cleared up.

1.       There is inconsistency in terms used.  In the abstract there are 6 terms used.  It is unclear if these are meant to be synonyms.  The terms are, courtship gender violence, dating violence, sexual abuse, intimate partner violence, gender violence and partner violence. 

In the introduction, GBV is used but it is not clear what this is an abbreviation for. 

I see on page 3, GBV means gender based violence (the term gender based dating violence is also used).  What is gender based violence?  How does it apply when dating partners are the same gender, would their violence be termed gender based?

Is  gender based violence is the same as dating violence. 

What is gender based dating violence and how is it different from dating violence? 

It seems that the authors are also viewing the problem they are studying as violence that is inflicted by men on their female partners.  This might be what gender-based violence is supposed to mean.  That needs to be overt throughout the paper.

Dating violence among teens is often reciprocal between partners.  They hurt each other and males can be victims too.  So this study isn’t really about dating violence and the title is misleading.

It should be clear to readers the position that the authors are coming from – this study is about gender-based dating violence which is understood to mean violence against women. 

What is gender-based violence?

I found this definition online: Gender-based violence and violence against women are two terms that are often used interchangeably, as most violence against women is inflicted (by men) for gender-based reasons, and gender-based violence affects women disproportionately. Council of Europe  If this is the definition that is underlying the use of this term, it needs to clearly stated.  As it is, there are places where its implied but as a reader, I find it confusing as to what your study is about.

2.       Are you using gender based violence and dating violence synonymously?  The first paragraph is stats on dating violence. Second paragraph switches to GBV. Please explain why its important to understand dating violence through a gendered lens.  Why are you using the term GBV instead of dating violence or intimate partner violence?  As I said above, it doesn’t seem like you are studying data violence. 

3.       Your intro includes assaults via electronic devices, GBV via technology, digital violence.  However, these terms are different than social network.  Do you mean social media apps like Instagram and snapchat?  To me social network refers to the people in one’s network not the apps or devices that we use to communicate virtually. So maybe the title should be social media and not social networking.  In the conclusion, the term social media is used, I think that term would work better throughout.

Also assault through electronic device is an awkward phrase assault connotes a physical presence.  I think abuse could work better than assault as a word choice.

“However, no attention has been paid to these facts.”  It is unclear which facts you are referring to. 

4.       Was data collected in Spanish?  If so, please explain the translation processes.

Include your interview guide in Spanish if you used one.  Was the term gender based violence defined in the your interviews and focus group?   How do they know that that term meant?

5.       Where were participants recruited from?  I don’t know what is meant by them being “collaborators.”  Did they all have history of experiencing violence?

6.       The purpose is “to describe and understand the perceptions and experiences of young Spaniards on intimate partner violence and its perpetration through social networks.” 

After the purpose, the term switches to gender based dating violence. However, these two terms are not interchangeable.

I see two ways to clarify, 1) adopt gender-based dating violence as your term and use it exclusively in the paper.  Explain to the reader what gender-based dating violence is and how its different than the term dating violence.  Change the language in the purpose.

2) adopt intimate partner violence as your term and use it exclusively throughout your paper. 

However, as you used the terms gender based violence and gender based dating violence in your interview questions, option 1 might be the best fix.  You will need to change your purpose.

Either way, choose one term and delete the other terms (except if needed for definitions) including courtship gender violence, dating violence, sexual abuse, gender violence, gender based violence, gender based dating violence, intimate partner violence and partner violence

7.       Results – once you have changed terms make sure the same terms are used in result section. Include a paragraph that explains your conceptual map, that map is hard to understand and too busy.  Also the central concept is maltreatment.  That is a new term that hasn't been used.  Switching terms is confusing to the reader.  The concept map should match the purpose - perceptions and experiences of GBV (or the term you choose) and its perpetuation through social media.  A paragraph explaining the map would be helpful to the reader.  And I think that the words "it is part of" "associated with" and "due to"  are beyond your data.  This is a small dataset that wasn't intended to evaluate the level or strength of relationships.  It would help with clarity to remove those words from the may.  Including them (if appropriate) within the explanatory paragraphy would be alot more helpful to the reader. 

Discussion: The first sentence states the purpose differently than was originally stated. Dating violence is not synonymous with gender based violence.  In dating violence people of all genders may be hurt and may also hurt others, often violence is reciprocal, especially among adolescents.

In the discussion, it was noted that participants viewed that violence isn’t only men hurting women.  That is an interesting belief (part of your purpose) so what does that mean for people in understanding how abuse is perpetuated through social media?  Does it mean we downplay violence and ignore the misogynist aspects?  Is all dating violence gender based?  Would it be helpful if we considered dating violence gender based? 

Comments on the Quality of English Language

Perhaps the term issue I have described is to due translation to English. Overall. the English is very good. 

Author Response

Dear reviewer,

I am enclosing the following cover letter to respond to the comments and improvements you have made to our research, thanking you in advance for the aspects analysed to improve the study. All changes made because of these have been highlighted in yellow in the article document.

First, regarding the variety of terms used to refer to gender-based dating violence throughout the article, we would like to apologise. This was due to an error in the translation from Spanish to English and has been modified so that the correct term (gender-based dating violence) appears where necessary throughout the study, as well as in the title of the study. Also, another term that has been changed throughout the article is "social networks" to "social media", following your advice, as well as the term "assault" to "abuse" through social networks, in the introduction section. Again, we apologise for these errors in translation.

As to why it is important to study dating violence from a gender perspective, it has been highlighted in the study in the following way as a result of your comments: "Due to a higher incidence in women, the fact that this type of violence tends to have worse consequences for the female gender and that research on gender-based violence has shown that these attitudes begin in courtship, it was considered important to analyze this type of violence with a gender-based approach (gender-based dating violence)". Related to this is your last point about this gender approach, as it has been noted in the study that some young people point out that this type of violence can be perpetrated both from boys to girls and vice versa. However, I consider the analysis through gender to be important since, as also observed in the study, despite this belief, most of the violence experienced by the participants has been from their boyfriends towards girls.

On the other hand, the translation process was carried out by a native translator who is an expert in the field. In addition, the interview and focus group guide in the original language, Spanish, has been added in Annex A. With respect to whether or not it was previously explained to the participants what gender-based dating violence was, the answer is negative. The aim was to inquire about the perceptions of these young people without interfering with their previous knowledge about this topic and thus be able to observe whether their conception of this type of violence varied.

Regarding the term "collaborators" in the inclusion criteria, we apologise for the error in translation. This term has been replaced by "person who cooperates" for better understanding.

In the discussion, we apologise for the change in the objective of the study. The correction has been made and we thank you for notifying us.

Finally, regarding the conceptual map, it has been considered convenient to eliminate the terms that linked the different "items" in order to help its understanding. An explanatory paragraph on the meaning of this figure has also been drafted for this purpose.

Reviewer 2 Report

Comments and Suggestions for Authors

See comments on attached word document.

Comments on the Quality of English Language

A thorough proof read is required to address the presentation and comprehension of the paper.

Author Response

Dear reviewer,

I am enclosing the following cover letter to respond to the comments and improvements you have made to our research, thanking you in advance for the aspects analysed to improve the study. All changes made because of these have been highlighted in yellow in the article document.

 First, we apologise for the confusion of terms due to the translation from English to Spanish. As a result, terms such as "social media" have been explained and "gender-based dating violence" has been taken over as the main term. We have also tried to change the phrase "free route for the exercise of gender-based dating violence" to make it more understandable and meaningful for the reader, using the phrase “Ease of engagement in dating violence”.

Regarding the title, we have considered that this new vision is more faithful to the topic of our study: Gender-based dating violence and social media among Spanish young people: a qualitative study.

About the summary of the article, we have not found any bibliographical references. If this is not the case, please inform us of the bibliographical reference for its subsequent elimination. 

On the other hand, we have added to the article several aspects that you have pointed out, in order to enrich it, such as the definition of courtship and its evolution throughout history, the reflection of the current digitalised society in young people's dating relationships, as well as an explanatory text of table 1 with the demographic information of the participants and the explanation of some of its terms.

In addition, the limitations of the study have been developed to address the level of university education of our participants.

Likewise, it was considered appropriate and beneficial to talk about educational programmes and the work of society and parents in combating gender-based dating violence in the discussion section of the study.

Regarding the ethical issues that arose during the interviews and the focus group, no participant was uncomfortable or reluctant to participate in the different topics that were discussed, showing a collaborative and participatory attitude.

Finally, we thank you for your recommendation to explore the benefits reported by participants about social networking in young people's dating relationships, and we will keep this in mind for future research.

Round 2

Reviewer 1 Report

Comments and Suggestions for Authors

Thank you for your work to revise this manuscript.  The revision has improved the quality of the paper. 

I have a few more suggestions for minor revision. 

Thank you for clarifying the connections between the concepts of your study.  Can you please add a definition of gender based violence?  I appreciate how you explained the connections between concepts and how you landed on gender based violence.  However, it is important to define the major concept of the paper.  Its ok if gender based violence is defined as violence directed towards women by men (if that is what the defintion is).  As a reader, I still am wondering about that term.  A definition would help. 

On page 3, the added highlighted content is useful.  However, this paragraph needs some revision.  It should all be in past tense and some of the wording is awkward.  Changing it to past tense is important to help the reader. 

On section 3.1, the participants voice a common but wron belief that its the woman's fault she's abused.  She's abused because she has now self-esteem and if her esteen was higher she could get out of the relationship.  It is important to share these statements in the results.  In the discussion, authors could add a line or two to point out that these views are common but are inaccurate.  Actually also somewhat contribute to perpetuation of violence. 

Second paragraph of section 3.1.1. needs to be revised.  Avoid use of you and your, its confusing to reader.  I would suggest a revision like this, "Participants identified that abusers limited their partners' freedom and self-espression by controlling their friendships, the clothes they wore, where they went and other decisions they made.  Participants also pointed out that abusers' aggressive behaviour and the use of high tones and insults were intended to devalue their victim and make them feel inferior."

Comments on the Quality of English Language

tense issues in methods and results

Author Response

Dear reviewer,

            I am enclosing the following cover letter in response to the comments and improvements you have made to our research, again thanking you for the points you have made to improve the study. All changes made as a result of these have been highlighted in yellow in the paper document.

We are sorry again for the errors in the translation from Spanish to English. Regarding the second paragraph of point 3.1.1, we have considered your proposed modification to be appropriate and correct for the better understanding of the reader. Thank you for your suggestion. We have also changed the paragraph on page 3 to the past tense to bring it into line with the rest of the article.

On the other hand, a description of the term "gender violence" has been added for a better understanding of the research topic.

Finally, a small paragraph has been added to the discussion to comment on the erroneous belief held by some of the participants that women are to blame for being abused, in relation to low self-esteem.

Thank you again for your comments and we hope that the changes made have enriched the article.